# The Effect of Silencing Fatty Acid Elongase 4 and 6 Genes on the Proliferation and Migration of Colorectal Cancer Cells

**DOI:** 10.3390/ijms242417615

**Published:** 2023-12-18

**Authors:** Aleksandra Czumaj, Jarosław Kobiela, Adriana Mika, Emmanouil Pappou, Tomasz Śledziński

**Affiliations:** 1Department of Pharmaceutical Biochemistry, Faculty of Pharmacy, Medical University of Gdansk, 80-211 Gdansk, Poland; adriana.mika@gumed.edu.pl (A.M.); tomasz.sledzinski@gumed.edu.pl (T.Ś.); 2Department of General, Endocrine and Transplant Surgery, Faculty of Medicine, Medical University of Gdansk, 80-211 Gdansk, Poland; jaroslaw.kobiela@gumed.edu.pl; 3Department of Surgical Oncology, Faculty of Medicine, Medical University of Gdansk, 80-211 Gdansk, Poland; 4Department of Surgery, Memorial Sloan Kettering Cancer Center, New York, NY 10065, USA; pappoue@mskcc.org

**Keywords:** colorectal cancer, lipid metabolism, fatty acid elongation, siRNA, cell proliferation, cell migration

## Abstract

Colorectal cancer (CRC) cells show some alterations in lipid metabolism, including an increased fatty acid elongation. This study was focused on investigating the effect of a small interfering RNA (siRNA)-mediated decrease in fatty acid elongation on CRC cells’ survival and migration. In our study, the elongase 4 (*ELOVL4*) and elongase 6 (*ELOVL6*) genes were observed to be highly overexpressed in both the CRC tissue obtained from patients and the CRC cells cultured in vitro (HT-29 and WiDr cell lines). The use of the siRNAs for *ELOVL4* and *ELOVL6* reduced cancer cell proliferation and migration rates. These findings indicate that the altered elongation process decreased the survival of CRC cells, and in the future, fatty acid elongases can be potentially good targets in novel CRC therapy.

## 1. Introduction

Colorectal cancer (CRC) is one of the most prevalent and lethal cancers worldwide [1,2]. There is growing evidence that reprogrammed lipid metabolism, including fatty acids (FA) metabolism, plays an important role in CRC tumor development and progression [3,4,5]. Studies, including ours, demonstrated an increase in the levels of very long-chain FAs (VLCFAs) in CRC patients’ serum and cancer tissue [6,7,8]. For the biosynthesis of very-long-chain saturated FA (VLC-SFA) and very-long-chain polyunsaturated FA (VLC-PUFA), elongation catalyzed by FA elongase 4 (ELOVL4) is essential.

FA elongase 6 (ELOVL6) catalyzes the elongation of saturated FA (SFA) and monounsaturated FA (MUFA) with 12–16 carbons to 18 carbons. It does not possess the capacity to elongate beyond that. However, it provides substrates for further elongation by other enzymes from the ELOVLs family [9,10,11]. The activity of ELOVL6 may determine the cellular level of stearate (C18:0), oleate (C18:1n − 9), and vaccinate (C18:1n − 7), all of which are important components of triglycerides, cholesterol esters, phospholipids, and sphingolipids. These derivates can be highly useful in cancer development as energy storage, membrane building blocks, and signaling molecules [12]. Moreover, FA elongated by ELOVL6 can be used as substrates for the synthesis of VLCFA [13].

The manipulation of the FA compositions by blocking elongase expression or activity could disrupt cancer cell maintenance. However, no well-established, commercially available chemical inhibitors exist for specific mammalians’ elongases. In 2009, the first inhibitor for ELOVL6 was obtained, and few compounds have been proposed since then. Despite that, proposed compounds were synthesized on-site by the research teams or selected based on in silico approaches [14,15,16]. Moreover, to the best of our knowledge, there are no chemical inhibitors for ELOVL4. Therefore, in this study, we focused on silencing *ELOVL4* and *ELOVL6* with small interfering RNA (siRNA) as a way to alter the FA elongation process. The siRNA has been extensively investigated as a potential molecular-based therapy for various cancer types. Several in vivo and in vitro studies have confirmed that siRNA-mediated silencing could significantly inhibit cancer cell proliferation [17,18,19]. Few studies reported the use of siRNA to decrease the expression of thymidylate synthase or a multidrug resistance gene, which resulted in increasing the sensitivity to chemotherapy in colon cancer in vitro models (HT-29 and RKO cell lines) [20,21]. The majority of CRC research has concentrated on genetic modifications in protein-coding genes. However, it should be mentioned that several studies have also analyzed the importance and correlation between CRC and miRNAs. MicroRNA levels can affect cell growth, proliferation, and metastasis in CRC [22,23,24,25].

The current study focused on the effect of silencing *ELOVL4* and *ELOVL6* genes on CRC cell proliferation and migration. This will help to evaluate if the FA elongation process can be a potential therapeutic target for CRC therapy. This is the first attempt to target ELOVL4 and ELOVL6 in CRC cell line models and normal colon cell models. Demonstrating that the elongation process would be a potential therapeutic target could initiate further research into the development of effective and selective elongase inhibitors.

## 2. Results

### 2.1. Baseline Expression of Elongase 4 and 6

Compared to normal colon mucosa tissue, CRC samples showed significantly elevated expression of ELOVL4 and ELOVL6 (Figure 1A). To confirm that we have chosen the right in vitro models, we evaluate the expression level of selected elongase genes in CCD-841-CoN, HT-29, and WiDr cell lines (Figure 1B).

Both selected CRC cell lines (HT-29 and WiDr) showed higher mRNA levels of ELOVL4 and ELOVL6 than normal colon cells (CCD-841-CoN). Changes in the expression levels of ELOVL4 and ELOVL6 observed in our in vitro model of colorectal adenocarcinoma corresponded with changes observed in patients’ CRC tissue. Interestingly, the HT-29 cell line seems to reflect the changes in the patients’ elongase expression levels more accurately in comparison to WiDr. The expression of ELOVL4 was three times higher in CRC patients than in normal tissue, whereas in cell models, it was three times higher in HT-29 and approximately six times higher in WiDr in comparison to normal colon cells. In the case of ELOVL6, the mRNA level was almost five times higher in CRC patients, whereas it was four times higher in HT-29 and seven times higher in WiDr compared to the control cells.

### 2.2. Modulation of ELOVL4 and ELOVL6 Genes Expression

siRNAs selected against ELOVL4 and ELOVL6 decreased the expression of the target mRNA with no significant knockdown effect on nontarget elongase mRNA levels (Figure 2A,B).

As shown in Figure 2A, siRNA *ELOVL4* was able to decrease the expression of ELOVL4 in all cell types, with the greatest effect in HT-29 (mRNA level lowered by 75%) and the lowest effect in normal colon cells (CCD-841-CoN), with the mRNA level lowered by 61%. Surprisingly, at the same time, in CCD-841-CoN cells, an increased level of *ELOVL2* mRNA and *ELOVL5* mRNA was observed. It is possible that in the absence of ELOVL4 in normal colon cells, there was some effect of replacement/compensation of function by other elongases. No such effect was observed for both CRC cell lines.

As shown in Figure 2B, siRNA *ELOVL6* was able to decrease the expression of *ELOVL6* in all types of cells, with the greatest effect on CCD-841-CoN (mRNA level lowered by 82%) and the lowest effect on WiDr (mRNA level lowered by 64%). The presence of siRNA *ELOVL6* did not change the expression level of any other elongase (ELOVL1–5), neither in normal colon cell lines nor in CRC cell lines.

As expected, positive control, negative control, and lipofectamine do not affect the expression levels of any elongases (*ELOVL1–6*) in all analyzed cell lines (Appendix A).

### 2.3. The Effect of ELOVL4 and ELOVL6 Silencing on Cell Viability

The obtained results illustrated that siRNA ELOVL4 significantly reduced the viability of CRC cells, with no statistically significant effect on normal colon cell lines (Figure 3A). The viability of CRC cell lines decreased by 80% and 60% in HT-29 and WiDr, respectively. The presence of siRNA *ELOVL6* significantly decreased the viability of all analyzed cells, cancerous as well as normal cell lines (Figure 3B). The viability of the cells decreased by 72%, 51%, and 38% in HT-29, WiDr, and CCD-841-CoN, respectively.

The sole presence of the siRNA construct (negative control) or the selected transfection agent (lipofectamine) does not affect the viability of any of the analyzed cell models (Appendix A). This confirmed that the observed effects are related to the specific effects of siRNA *ELOVL4* and siRNA *ELOVL6*. The loss in cell viability with siRNA *GAPDH* confirms the high transfection efficiency (Appendix A).

### 2.4. The Effect of ELOVL4 and ELOVL6 Silencing on Cell Migration

To understand if FA elongase silencing could influence cell mobility, a migration assay was carried out in control CCD-841-CoN colon cells and in CRC HT29 and WiDr cells (Figure 4A–C, respectively). As illustrated in Figure 4A, in normal colon cells, the migration ability is significantly decreased in the presence of siRNA *ELOVL4* or siRNA *ELOVL6* after 24 h. However, after 48 h and 72 h, there was no statistically significant difference between the migration rate of cells in control conditions and cells incubated with siRNA *ELOVL4*. In the presence of siRNA *ELOVL6*, the migration rate decreased in CCD-841-CoN cells in all analyzed time points.

In both CRC cell lines, the presence of siRNA *ELOVL4* and siRNA *ELOVL6* statistically significantly decreased the migration rate in all analyzed time points (Figure 4B,C).

The sole presence of the siRNA construct (negative control) or the selected transfection agent (lipofectamine) does not affect the mobility of the analyzed cells (Appendix A). This confirmed that observed effects are related to the presence of specific siRNAs–siRNA *ELOVL4* and siRNA *ELOVL6.*

## 3. Discussion

To investigate if ELOV4 and ELOVL6 are potential therapeutic targets in CRC treatment, the present study was designed to evaluate the effect of *ELOVL4* and *ELOVL6* knockdown on CRC cell viability and migration capacity.

Our findings revealed that expression of *ELOVL4* and *ELOVL6* is elevated in CRC tissues compared with normal tissues, which is consistent with our previous reports [8,26], as well as in HT-29 and WiDr compared to the normal colon cell line CCD-884-CoN. This is the first attempt to compare the expression of the enzymes catalyzing the FA elongation process among CRC cell lines. Few studies compare epigenetic and genetic features of selected CRC cell lines, but none of them included FA/lipid metabolism genes in the analysis [27,28,29].

We found that the downregulation of *ELOVL6* by siRNA decreased all analyzed cell lines’ viability and migration rate, including normal colon cells. In contrast, the downregulation of *ELOVL4* decreased the viability and migration rate of CRC cells, but importantly, it had no effect on normal colon cells. Together, these data suggest that ELOVL4 may be potentially a better target in CRC treatment. Until today, there is no data on the effect of siRNA-based inhibition of the FA elongation process on CRC cells. A few studies on the modulation of *ELOVL6* expression have focused mainly on cell FA composition. After the knockdown of *ELOVL6* decreased the level of C18:0, C18:1n − 9, an increased level of C16:1n − 7 was observed in the skeletal muscle cell line and insulinoma cell line [30,31]. There is some evidence that the knockdown of *ELOVL6* inhibits migrations of some cells, like normal vascular smooth muscle cells [32] or glioblastoma multiforme cells [33]. However, Zakharova et al., demonstrated that inhibition of ELOVL6 does not change the migration activity of breast cancer cells (cell lines MCF7 and MDA-MB-231) [34].

The current study showed that the downregulation of *ELOVL4* was associated with a higher expression level of *ELOVL2* and *ELOVL5* in normal colon cells. No such findings have been observed after siRNA *ELOVL6* treatment. This may explain the differences in the consequences of *ELOVL4* versus *ELOVL6* downregulation. Similar to ELOVL4, ELOVL2 and ELOVL5 are involved in PUFA metabolism [35,36]. Although ELOVL2 preferentially elongates C22 PUFA up to C30 of the n − 6 series, ELOVL4 preferentially elongates C26 PUFA up to C36 of the n − 3 series, and ELOVL5 preferentially elongates C18 PUFA up to C22 PUFA, it is possible that a certain range of substrates can be processed by more than one elongase [37,38]. For example, in mammals, conversion 22:4, n − 6 into 24:4, n − 6 and 22:5, n − 3 into 24:5, n − 3 can be catalyzed by ELOVL2 or ELOVL4 [39,40]. Moreover, the compensation of the elongase function was observed in some fish species, where elongase 4 may be able to compensate for the lack of elongase 2, whereas the knockdown of elongase 5 resulted in the increased expression of elongases 2 and 4 [41,42,43]. However, data from mammalian studies are very limited. There is one study where in the human prostate cancer cell line (LNCaP) with the knockdown of *ELOVL2*, *ELOVL5*, or *ELOVL7*, no compensatory increases in the expression of the other *ELOVLs* were observed in the cancer cell line, and no normal cell line was analyzed [44].

The present research, for the first time, demonstrates that *ELOVL4* and *ELOVL6* knockdown could effectively diminish the proliferation and migration rate of CRC cells. Nevertheless, taking into consideration that the knockdown of *ELOVL6* also has a significant effect on normal colon cells, whereas the silencing of *ELOVL4* specifically decreases the proliferation of CRC cells without affecting normal colon cells, targeting *ELOVL4* appears to be more promising as a potential CRC therapy. Undoubtedly, the strengths of this research are the inclusion of two different in vitro CRC models and the inclusion of the normal colon cell line.

## 4. Materials and Methods

### 4.1. Tissue Samples

The study includes 88 tissue samples obtained after surgical resection of a large bowel segment with stage I to IV CRC. Written informed consent was obtained from all the patients prior to this study. Two tissue samples were obtained from each patient: the cancer sample and the normal colon mucosa sample (tissue at least 5 cm from the tumor interface). All samples were frozen in liquid nitrogen immediately after collection and stored in aliquots at −80 °C until the analysis. The protocol of this study was compliant with the Declaration of Helsinki of the World Medical Association and with approval from the Local Bioethics Committee at the Medical University of Gdansk (decision no. NKBN/487/2015).

### 4.2. Cell Culture

HT-29 and WiDr cell lines were used as the colorectal adenocarcinoma model, and CCD-841-CoN was selected as a normal human colon model. All cell lines were obtained from ATCC (American Type Culture Collection, Manassas, VA, USA) maintained in basic media recommended by the supplier supplemented with 10% fetal bovine serum, 100 units/mL penicillin, and 100 μg/mL streptomycin at 37 °C in a humidified atmosphere of 5% CO_2_. Cells were seeded on 24-well culture plates at seeding density 0.05 × 10^6^ for gene expression experiments. For the viability test, 0.01 × 10^6^ cells/well were seeded on 96-well cell culture plates (Sarstedt, Nümbrecht, Germany). For cell migration assay, cells were seeded at a density of 3 × 10^5^ cells per well in 24-well cell culture plates (Corning Costar, Corning, NY, USA).

### 4.3. Modulation of Fatty Acid Elongases Expression with siRNA

Negative control, positive control, ELOVL4, and ELOVL6 siRNA were obtained from Thermo Scientific (Waltham, MA, USA). For the infection of HT-29, WiDr, and CCD 841 CoN with Elovl-4 siRNA, Elovl-6 siRNA, or negative control siRNA cells were seeded onto selected experimental plates. After reaching 60% confluency, the basic culture medium was replaced with Opti-MEM medium (base medium) with Lipofectamine RNAiMAX and selected siRNA in a 1:1 ratio. The final siRNA concentration was five pmol per well. Cells were incubated with selected siRNA for 72 h. To optimize siRNA delivery conditions and confirm high levels of delivery in each experiment, siRNA GAPDH was used as a positive control. A non-targeting siRNA was used as negative control (siRNA(-)) to detect any possible non-specific effects of siRNA delivery.

### 4.4. Cell Proliferation Assay

The level of cell proliferation after siRNA infection was analyzed by MTT test and was based on counting the number of viable cells with the Synergy HT Microplate Reader (BioTek, Winooski, VT, USA). After 72 h in experimental conditions, 10 uL of 3-(4,5-dimethylthiazol-2-yl)-2,5-diphenyltetrazolium bromide (MTT) solution (5 mg/mL) was added into each well and incubated with the cells for four h in a CO_2_ incubator to allow the transformation of MTT dye to formazan salt. Then, MTT dyes were removed, and 150 μL DMSO was added to the wells. Plates were analyzed by the colorimetric method at a wavelength of 590 nm.

### 4.5. Cell Migration Assay

For cell migration assay, cells were seeded at a density of 3 × 10^5^ cells per well in a 24-well and allowed to grow to each 90–100% confluency. Then, a scratch was created on monolayer cells using a yellow sterile pipette tip (2–200 µL, Eppendorf, Hamburg, Germany). Afterward, the cells were treated with siRNA *ELOVL4*, siRNA *ELOVL6*, siRNA *GAPDH*, siRNA(-), or lipofectamine. The created gap was photographed at 0 h, 24 h, 48 h, and 72 h using EVOS XL Core Imagine System (Thermo Scientific). The migration rate of the cells through the gap area was calculated as a percentage of the cell-covered area.

### 4.6. RNA Extraction and Real-Time Polymerase Chain Reaction (PCR) Analysis

Using the standard procedure, the total RNA was extracted from cells and frozen tissues with the GenElute Mammalian Total RNA Miniprep Kit (Merc, Darmstadt, Germany). The quantity and quality of RNA prior to the downstream experiment were assessed using the Experion Automated Electrophoresis System (Bio-Rad, Hercules, CA, USA). According to the manufacturer’s instructions, the extracted RNA was used to synthesize the cDNA using the RevertAid First Strand cDNA Synthesis Kit (Thermo Scientific). For real-time PCR analysis, cyclophilin A was used as a reference gene. The relative gene expression level of the *ELOVL1*–*6* genes in the presence or absence of selected siRNA was calculated using the 2^−ΔΔCT^ analysis method.

### 4.7. Statistical Analysis

All data are presented as mean ± standard deviation (SD) of three independent experiments done in triplicates. Before the start of the experiment, we estimated the required minimum number of tissue samples by the “A priori required sample size” test by G*Power Software 3.1.9.7 [45]. The statistical significance of the observed differences in the study parameters was verified with a two-tailed Student’s *t*-test (cell line experiments) and paired samples Student’s *t*-test (tissue experiment). For more than two groups for comparison, we used ANOVA with a post hoc Scheffe test. *p* < 0.05 was considered statistically significant. All statistical calculations were carried out with Statistica 13 (TIBCO, Palo Alto, CA, USA).

## 5. Conclusions

In conclusion, our results suggest that the siRNA-based inhibition of FA elongation could effectively diminish the proliferation and migration of CRC cells. This study shows for the first time that ELOV 4 and ELOVL6 knockdown effectively reduces the proliferation and migration of CRC cells. In particular, ELOVL4 seems to be a specific target in CRC cells since its silencing in normal colon cells did not influence their viability, whereas ELOVL6 knockdown also significantly affects normal colon cells. This could be regarded as a potential strategy to develop novel approaches for the improvement of CRC therapy. However, there is a need for more in vitro and in vivo studies to further validate the value of the siRNA-based modulation of FA elongation as a beneficial CRC treatment strategy and for studies dedicated to the discovery of elongase inhibitors.

## Figures and Tables

**Figure 1 ijms-24-17615-f001:**
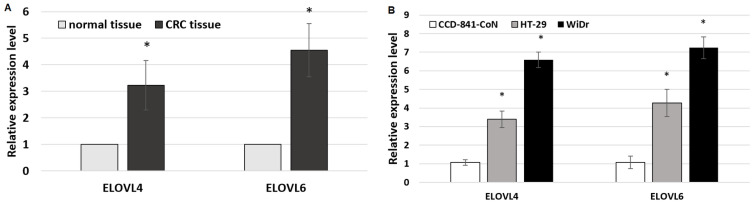
(**A**) ELOVL4 and ELOVL6 mRNA levels in normal colon and CRC tissue. Data are presented as mean ± SD. * *p* < 0.05 compared to normal tissue. (**B**) ELOVL4 and ELOVL6 mRNA levels in selected cell lines. Data are presented as mean ± SD. * *p* < 0.05 compared to CCD-841-CoN.

**Figure 2 ijms-24-17615-f002:**
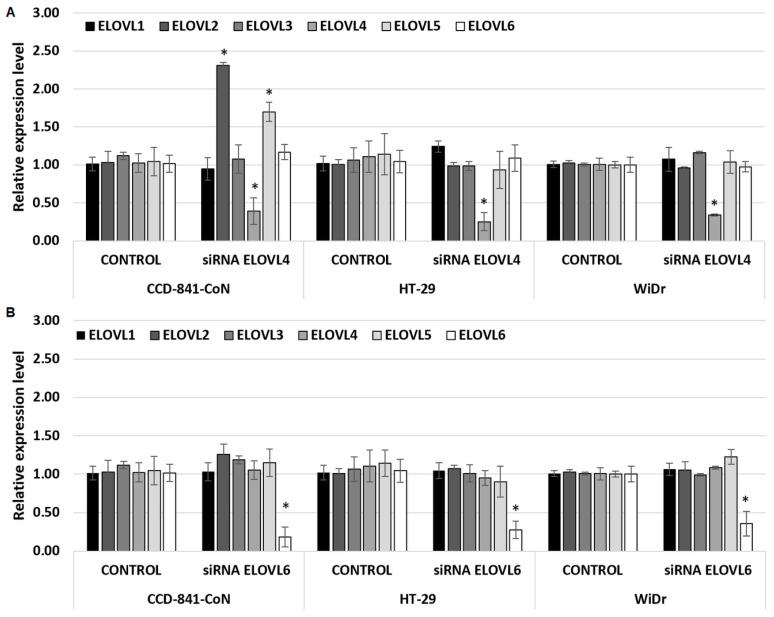
Relative expression levels of elongases 1–6 (ELOVL1, ELOVL2, ELOVL3, ELOVL4, ELOVL5, and ELOVL6) in control conditions (base medium) and after 72 h with siRNA ELOVL4 (**A**) or siRNA ELOVL6 (**B**). Data presented as mean ± SD. * *p* < 0.05 compared to the control condition (base medium).

**Figure 3 ijms-24-17615-f003:**
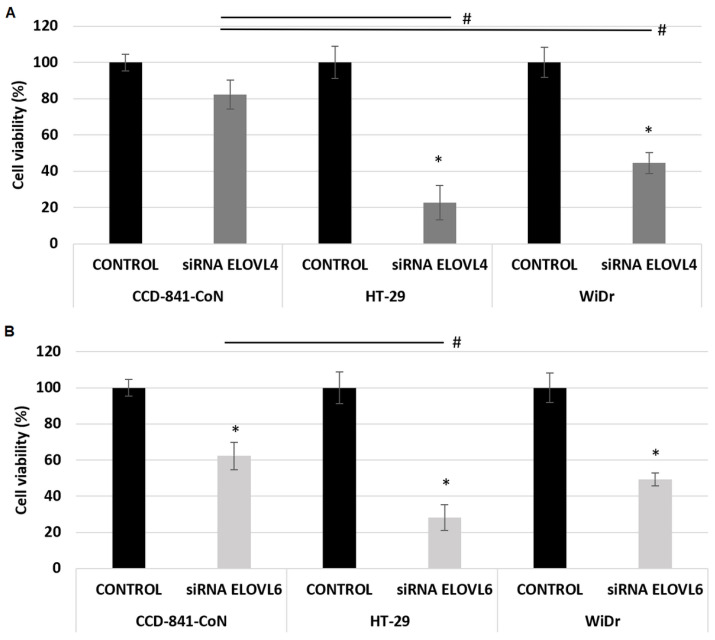
Cell viability (%) after 72 h incubations with siRNA *ELOVL4* (**A**) and siRNA *ELOVL6* (**B**). Data presented as mean ± SD. * *p* < 0.05 compared to the control condition (base medium), # *p* < 0.05 ANOVA.

**Figure 4 ijms-24-17615-f004:**
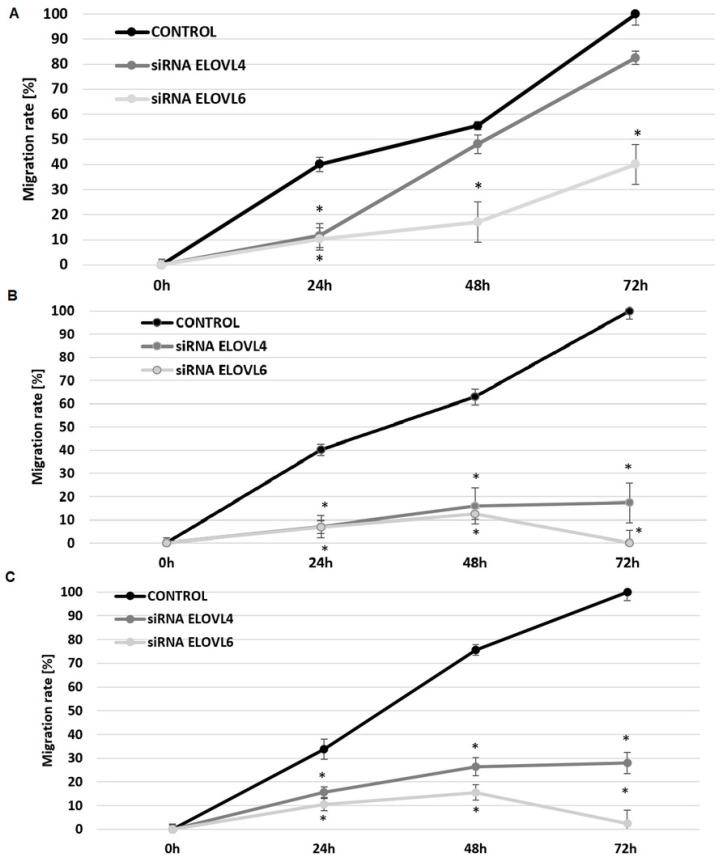
The migration rate of CCD-841-CoN (**A**), HT-29 (**B**), and WiDr (**C**) at 0 h, 24 h, 48 h, and 72 h incubation with siRNA *ELOVL4* or siRNA *ELOVL6.* * *p* < 0.05 compared to the control condition (base medium).

## Data Availability

All study data can be viewed in this manuscript.

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
