# Peer review of "The Effect of Silencing Fatty Acid Elongase 4 and 6 Genes on the Proliferation and Migration of Colorectal Cancer Cells"

_ijms, 2023, doi:10.3390/ijms242417615_

Round 1
Reviewer 1 Report
Comments and Suggestions for Authors
3 December 2023
Ms. Ref. No.: ijms-2766788
Journal: International Journal of Molecular Sciences.
Title: The effect of silencing fatty acid elongase 4 and 6 genes on proliferation and migration of colorectal cancer cells
Comments:
Thank you for your efforts in writing this article on a very pertinent topic. I have some observations where mentioned in the following paragraphs that will be useful for its improvement:
1- In this study was conducted by 88 tissue samples, how calculate this sample size?
2- How much is the association between the stage number of colorectal cancer CRC and fatty acid elongation?
3- In the step of Modulation of fatty acid elongases expression with siRNA, why incubation was done for 72h?
4- In the section 2.1. Baseline expression of elongase 4 .And it was mentioned that 29 cell line seems to reflect the changes, what about others? (Figure 1)
5- According to Figure 3, Is the concentration important on cell viability and has any effect?
6- Additionally, some of the Following reference can be included in the introduction part for more readability:
· https://doi.org/10.3390/ijms242317084
· https://doi.org/10.1111/ahg.12443
· https://doi.org/10.1016/j.genrep.2022.101578
· https://doi.org/10.3390/ijms242316740
Author Response
RE: The authors would like to express their gratitude for the time spent reviewing our manuscript
1- In this study was conducted by 88 tissue samples, how calculate this sample size?
Re: We included all tissue samples available at the time of performing this experiment that met the inclusion criterion. Before the start of the experiment, we estimated the required minimum number of tissue samples by the “A priori required sample size” test by G*Power Software. Information added to the manuscript (line 256-258).
2- How much is the association between the stage number of colorectal cancer CRC and fatty acid elongation?
Re: To the authors` knowledge data on changes in fatty acid profile at each CRC stage are very limited. Some studies have shown that elevated levels of total cholesterol, triglycerides, and LDL are associated with a higher CRC stage (e.g. DOI: 10.2147/RMHP.S260113). Our previous work has shown that very long FAs were the most highly increased FAs in cancer tissue compared to normal colon mucosa (doi: 10.3389/fonc.2021.689701). However, due to a limited number of age- and sex-matched samples from each CRC stage, we were not able to analyse each stage separately in regards of fatty acid elongases activity. However, Liu et al. have shown that cholesteryl ester with long fatty acids increased in stage III/IV CRC in comparison to stage I/II CRC and can separate early-stage CRC (I/II CRC) patients from advanced-stage CRC (III/IV) patients with high sensitivity and specify (doi:10.1007/s00216-019-01872-5).
3- In the step of Modulation of fatty acid elongases expression with siRNA, why incubation was done for 72h?
Re: The time was selected on the basis of previously conducted optimisation experiments. The manufacturer of the transfection agent states that transfection may take 1 to 3 days. We have tested each of these time points (24, 48, and 72h) and we selected the time point with the highest silencing efficiency. We also took into consideration the visible effect on cell viability.
4- In the section 2.1. Baseline expression of elongase 4 .And it was mentioned that 29 cell line seems to reflect the changes, what about others? (Figure 1)
Re: Both analysed CRC cell lines reflect the nature of the change observed in the tissues. The HT-29 cell line appears to more faithfully capture the variations in the patients' elongase expression levels when compared to WiDr. In CRC patients, the expression of ELOVL4 was three times greater than in normal tissue; in cell models, it was roughly six times higher in WiDr compared to normal colon cells, and three times higher in HT-29. When compared to normal cells, the mRNA level for ELOVL6 was nearly five times higher in CRC patients, four times higher in HT-29, and seven times higher in WiDr. (lines 75-85). In terms of other cell lines, according to “The Human Protein Atlas” similar levels of ELOV6 transcripts to HT-29 have LS180 and COLO206. The literature lacks of more extensive data on the expression of ELOVL4 in different CRC lines. Regarding other genes, from our other studies, we know that selected CRC cell lines reflect changes in expression level in many lipid metabolism genes, e.g. ACAT, ACLY, DGAT, FASN, SCD1, HMGCR
5- According to Figure 3, Is the concentration important on cell viability and has any effect?
Re: Concentration of the cell is important for cell viability tests. If the seeding density is too high, the cells might already reach overconfluence after a very short time. This leads to impaired proliferation and cell detachment from the surface of the plates, which results in non-reproducible experimental conditions. The concentration of transfection agent (siRNA + lipofectamine) is also important for cell viability. The concentration of siRNA determines the quantity of lipofectamine. It is well-documented that lipofectamine may exhibit cell toxicity at higher concentrations. In pre-experimental optimisation tests, we investigated the behaviour of our selected cell lines at different concentrations of lipofectamine to ensure that the observed changes were only related to the presence of siRNA.
6- Additionally, some of the Following reference can be included in the introduction part for more readability:
- https://doi.org/10.3390/ijms242317084
- https://doi.org/10.1111/ahg.12443
- https://doi.org/10.1016/j.genrep.2022.101578
- https://doi.org/10.3390/ijms242316740
RE: Thank you for your suggestions, after analysing we have taken some of them into account to increase the readability of the introduction (lines: 55-59).
Reviewer 2 Report
Comments and Suggestions for Authors
Over the past few years, fatty acids has recently attracted a lot of attention. In my opinion, this is a well-planned study and has a series of experiments that build on the fundamental aspects. Overall, I think this work is publishable with few minor comments.
In introduction section: author should add more discussion about the importance and significance of this study.
In all the figures, please increase the quality resolution.
Reason for choosing this particular cancer cells?
In order to improve the quality of your paper, I would appreciate it if you could check the fluorescence microscopy for Control, ELOVL4 and ELOVL6 if it is possible.
In conclusion section or end of the manuscript: it is suggested to add a summary of the research including significance findings and also some qualitative results that could enhance the readability of this research
Author Response
Over the past few years, fatty acids has recently attracted a lot of attention. In my opinion, this is a well-planned study and has a series of experiments that build on the fundamental aspects. Overall, I think this work is publishable with few minor comments.
RE: The authors would like to express their gratitude for the time spent reviewing our manuscript
1) In introduction section: author should add more discussion about the importance and significance of this study.
RE: Information about the importance and significance of this study was added to the Introduction section (lines: 62-65).
2) In all the figures, please increase the quality resolution.
RE: All figures were improved to have a better quality resolution (at least 300dpi).
3) Reason for choosing this particular cancer cells?
RE: There are over 30 colon cancer cell models, based on the literature review and our experience in handling cell culture we selected HT-29, WiDr, and as a control cells CCD-841-CoN. Moreover, HT-29 is widely used in cancer research as a colon cancer model to study various aspects of cancer biology, including cell proliferation, differentiation, and drug resistance. The use of WiDr is a less common CRC model in studies, but the addition of a second CRC line to our experiment allows preliminary conclusions to be drawn that the observed effect will apply to CRC in general and not to a single cell line. In any experiment with an abnormal cell line, we implemented an equivalent normal cell line that can be used as a control (CCD-841-CoN cell line). HT-29 is a cell line with epithelial morphology that was isolated in 1964 from a primary tumour obtained from a 44-year-old, White, female patient with colorectal adenocarcinoma. WiDr is a cell line exhibiting epithelial morphology that was isolated from the colon of a 78-year-old female patient with colorectal adenocarcinoma cancer. CCD-841-CoN cells were isolated from normal human colon tissue of female in 21 week of gestation.
4) In order to improve the quality of your paper, I would appreciate it if you could check the fluorescence microscopy for Control, ELOVL4 and ELOVL6 if it is possible.
RE: Unfortunately authors do not currently have access to a fluorescence microscope. Our experiment was preliminary and, due to the limited number of results, we decided to publish it as a "Communication". In the continuation of our research, we will certainly keep this suggestion in mind. Some publications confirmed that observed mRNA level of ELOVL4 and ELOVL6 is in line with protein level (e.g. DOI:10.2337/db09-0728, doi:10.1152/ajpendo.00544.2010)